# Comparative Study on Interface Fracture of 4th Generation 3-Steps Adhesive and 7th Generation Universal Adhesive

**DOI:** 10.3390/ma16175834

**Published:** 2023-08-25

**Authors:** Ștefan George Călinoiu, Cornelia Bîcleșanu, Anamaria Florescu, Dan Ioan Stoia, Cătălin Dumitru, Marian Miculescu

**Affiliations:** 1Doctoral School of Dentistry, Organizing Institution of University Doctoral Studies, “Titu Maiorescu” University, 67A Gh. Petrascu Street, 040441 Bucharest, Romania; calinoiu_stefan@yahoo.com; 2Faculty of Dental Medicine, “Titu Maiorescu” University, 67A Gh. Petrascu Street, 040441 Bucharest, Romania; corneliabicle@yahoo.com (C.B.); dumitru.catalin76@gmail.com (C.D.); 3Department of Mechanics and Strength of Materials, Faculty of Mechanical Engineering, Politehnica University of Timisoara, No.1 Mihai Viteazu Avenue, 300222 Timisoara, Romania; 4Faculty of Material Science and Engineering, Politehnica University of Bucharest, 313 Splaiul Independentei, District 6, 060042 Bucharest, Romania; marian.miculescu@upb.ro

**Keywords:** dental adhesives, interface, FEM analysis, SEM investigation, crack propagation, SIFs

## Abstract

The purpose of this paper is to compare the fracture behavior of interfaces obtained using fourth-generation and universal dental adhesives. The study relies on optic and SEM to evaluate the dentin–adhesive–restoration material interface of the samples and also on FEA simulation of fracture behavior. Specimen fabrication relied on 20 extracted teeth, in which class I cavities were created according to a protocol established based on the rules of minimally invasive therapy. For the direct adhesive technique, the adhesives used were: three-step All Bond, three-batch A and one-step Clearfil Universal Bond Quick-batch B. The restoration was performed with the same composite for both adhesives: Gradia direct posterior. The simulation used a 3D reconstructed molar on which geometric operations were performed to obtain an assembly that replicated a physical specimen. Material properties were applied to each component based on the information found in the literature. A simplified model for crack propagation was constructed, and using the fracture mechanics tool in Ansys 2019, the stress intensity factors that act at the crack tip of the adhesive interface were obtained. Mechanical simulation and microscopic investigation showed us how the interface of the dentine–adhesive–filling material performed in cases of both dental adhesives and for a certain loading condition. Important differences were identified among the adhesives, the fourth generation being superior to the fourth generation especially due to the separate steps in which the tooth surface was prepared for adhesion.

## 1. Introduction

Dental adhesives allow clinicians to achieve a more conservative cavity design, preserving healthy tissue since mechanical retention is no longer necessary. The most likely causes of secondary caries are micro-infiltrations, which represent a major problem. Using adhesives, restorations are not only more aesthetically pleasing but also create an intimate seal between healthy tissue and the restoration, thus reducing or eliminating the risk of secondary caries.

Dr. Michael Buonocore first introduced dental adhesives in 1955 and discovered the benefits of acid etching to improve enamel adhesion. Advanced technologies and dental adhesives have evolved from no-etch to full-etch (fourth and fifth generations) and to self-etching (sixth, seventh and eighth). The latest innovations in adhesives include universal adhesives that can be used as self-etching [1].

The fourth generation of adhesives was created between the 1980s and 1990s and provided the first adhesives that achieved complete removal of the stain layer. In this generation, three primary components were used: acid, primer and adhesive. According to the protocol, each component was individually packaged and applied sequentially. Compared to previous generations, the hybrid layer consisted of the resin-infiltrated surface layer on the dentin and enamel, providing high bond strengths and a dentin seal with significantly reduced marginal leakage [2].

The marginal integrity of dental adhesion was investigated using different adhesive strategies. The self-etch and selective enamel etch strategies were used in two application periods on primary molars. The authors found that different adhesive strategies provided no statistical difference, while the application time within the same adhesive was significantly different. Therefore, the universal adhesives used to restore class-II cavities, applied either in selective mode or in self-etch mode, resulted in comparable marginal integrities [3].

Some authors have investigated the impact of artificial aging of four nanocomposite resins on color stability and hardness properties. All investigated parameters demonstrated statistically significant results between samples before and after aging [4].

An extensive study of six universal adhesives was conducted on sound and caries-affected dentin after 18 months to reveal the antimicrobial activity, cytotoxicity, resin–dentine microtensile bond strength and nanoleakage. Poor performance in terms of microtensile bond strength and nanoleakage was observed when the adhesives were applied in caries-affected dentin, mainly after 18 months. Additionally, not all universal adhesives behaved the same in terms of antimicrobial activity and cytotoxicity [5].

The bonding properties of three two-step etch-and-rinse adhesives (2-ERAs) to those of three universal adhesives (UAs) applied with an etch-and-rinse strategy (ER) to dentin were determined directly and after one year of water storage. The results revealed that microtensile bond strength and nanoleakage properties are better for universal adhesives after one year of water storage. The authors concluded that the use of UAs applied with the ER strategy seemed to be a more effective technique for maintaining adhesion to dentin substrate than 2-ERAS [6].

The microtensile bond strength and nano-leakage properties immediately before and after 6 months of aging were determined using three types of adhesive systems in relation to the dentin. One adhesive was used in two steps (primer and bonding), while the other two adhesives were used also in two steps (phosphoric acid + universal adhesive) but also in one step by applying the universal adhesive directly. The tissue base consisted of dentin surfaces from human third molars. The results showed superior adhesion in the case of acid etching used as an independent step [7].

AllBond3 (Bisco), the fourth generation adhesive used in this study, offers micro-mechanical adhesion to all types of substrates, being compatible with all materials: photopolymerizable, self-solidifying or dual. The most important feature of this adhesive is its hydrophobicity, which allows it to have a very long intra-oral lifetime compared to other adhesive systems. It is a built bicomponent and is radio-opaque, thus being visible on dental X-rays [8].

Clearfil is a universal one-component photopolymerizable adhesive used in direct and indirect restorations in combination with total and selective surface conditioning techniques. These techniques are directly proportional to the degree of conditioning of the surfaces, as well as to the mode of diffusion of the cement components in the biological structure [9].

The restoration materials widely used in dentistry are resin-based composites (RBCs) and glass ionomer cements (GICs) [10]. The RBCs are mainly used for posterior teeth restorations, in which treatment of conservative cavity preparation can be performed, with very good results in longevity when restoring class I and II cavities. It is mercury free; it can match the color of the natural tooth, and it requires a compatible adhesive for bonding to teeth [11]. GICs appear as an alternative to amalgam restorations and the inclusion of reinforcements (zinc- and resin-modified glass ionomers) make this class of materials excellent for restorations, mainly because of their capacity for fluoride release, minimal shrinkage at curing and strong dentin bond [10,12].

Gradia composite resin is based on a co-monomer matrix and a pre-polymer resin reinforcing agent. It is light-curable and offers very good quality in reconstruction aesthetics, degree of polish, wear resistance and good tear toughness. Elastic bending energy determines the flexible behavior of the resin compared to other, more rigid components [13].

From the clinical use point of view, both adhesives have the same type of indications. Despite performance, dentists require a simplified system; thus, universal adhesives were developed. This new generation of self-etching agents has acidic hydrophilic monomers and can be easily used on etched enamel after saliva or moisture contamination.

Mechanical testing of dental materials has been performed by many authors. Some approaches have used numerical simulations, in which the stress and strain states of a reconstructed dental model are subjected to physiological loading to determine the mechanical behavior of the structure [14,15,16,17]. Other authors have investigated the surface morphology and fracture interfaces by microscopy and optical coherence tomography to determine the adhesion of the materials to dental substrates [18,19,20,21]. Nevertheless, the fracture mechanics on dental materials have been less investigated.

Modes I and II of failure of composite resins was investigated by Hidehiko Watanabe et al. using the Brazilian disc method. Twenty-five specimens of each material were subjected to compression on a Zwick machine at a reduced loading rate to allow for crack propagation. The stress intensity factor was then calculated, and based on the results, the averages and standard deviations of the fracture toughness in modes I and II (K_IC_ și K_IIC_) were determined [22].

Using three testing methods, namely tensile, three-point bending and four-point bending, Ayse Mese et al. studied the mode I toughness behavior of seven commercially available dental resins. Based on the experimental results, the correlation coefficients were calculated, as well as the characteristic average values of fracture toughness for each individual test. The highest fracture toughness values of dental resins were obtained in four-point bending tests [23].

The fracture mechanics of dental adhesives were determined on notched short bar specimens from a batch of 48 specimens by Kimberly Howard et al. In addition to determining the fracture toughness parameters in mode I, the study highlighted the incidence of fracturing at the dentin–adhesive interface. The study showed that more than 50% of the samples studied failed by crack propagation at this interface, and only 8% failed by cracking the dental composite. The way that dental adhesive and the composite remain connected while the adhesive is pulled from the dentin tuberosity as a result of crack propagation was presented in the paper. It was found that the fracture toughness values are not very different depending on the type of adhesive but were significantly lower than those of ceramics and dental composites. This outcome indicates brittle failure at the adhesive–dentin or adhesive–ceramic interface [24].

The main concern related to the fracture toughness of dental restorative composites is to increase the value of this property and consequently to extend the life of the dental construct. However, the complexity of the forces acting in the oral cavity makes it very complicated to find a suitable method for determining the toughness of composite resins. Dental restorations are often subjected to continuous mechanical stresses, leading to progressive degradation and crack propagation over time. Moreover, vacancies in the material structure, improperly created interfaces or residual stresses of any of the materials involved worsen the medium- and long-term performance of restorative constructions [25,26,27,28].

Th purpose of this study was to evaluate the interface behavior of dental restorations using two dental adhesives: a three-steps dental adhesive of fourth generation and a one-step universal adhesive of seventh generation. To reveal the differences and similarities between them, microscopical evaluation of the restored samples and finite element analysis (FEA) of the simulated interface fractures were conducted.

## 2. Materials and Methods

The investigation protocol (Figure 1) consisted of two stages. One refers to the microscopical evaluation of the interfaces obtained by the restoration of natural teeth using two adhesives: one universal and one belonging to the fourth generation; while the other investigation refers to the numerical simulation of the restored interface, relying on fracture properties of the dental materials collected from the literature.

### 2.1. Specimen Fabrication

Restoration was performed on 20 extracted teeth. After extraction, they were cleaned and kept in distilled water until their use. The extracted teeth were integrated into artificial arches and divided into two groups of 10 teeth each. Class I cavities were created in each tooth that affected the enamel and dentin, according to a protocol established according to the rules of minimally invasive therapy.

For the direct adhesive technique, the adhesives used were: three-step All Bond 3 (Bisco Inc., Schaumburg, IL, USA)—batch A and direct Clearfil Universal Bond Quick (Kuraray Noritake Dental Inc., Kurashiki, Japan)—batch B. The restoration was performed with the same composite for both adhesives, Gradia Direct Posterior (GC). We selected Gradia Direct Posterior due to the unique UDMA technology, which offers minimal shrinkage during photopolymerization.

The composite resin Gradia Direct Posterior is a light-cured, radio-opaque micro-filled hybrid composite resin, selected because of its low polymerization shrinkage stress (1.5 MPa), which reduces latent energy on bonding interfaces. The reduced interface stress has clinical benefits, such as protection of marginal integrity, thereby avoiding microleakage/secondary decay, and decreased postoperative sensitivity. It contains urethane dimethacrylate (UDMA) 20–25%, bismethacrylate 1–5%, dimethacrylate 1–5%, silica and pre-polymerized fillers. Additionally, according to the producer, it shows good fracture toughness to resist occlusal stress, a low modulus of elasticity, resistance to bending and occlusal forces, as well as the spread of cracks. From a clinical point of view, it shows good manufacturability, high wear resistance and low wear on opposing dentition [13].

The cold incorporation of the samples was performed in epoxy resin that was poured into a cylindrical mold and mixed with the hardener to induce the exothermic hardening reaction. An important property of the resin is polymerization shrinkage, which can be high for non-epoxy resins. The disadvantage of resin shrinkage is the cleavage that can occur at the resin–sample interface. To prevent the extraction of the hardened resin from the mold, an agent to prevent adhesion was sprayed on the walls prior to casting. Small clips were used to support the samples and prevent floating inside the mold during the incorporation stage (Figure 2).

The sample preparation ended with several steps that were accomplished after incorporation and that are mandatory for microscopic investigation: sample sectioning for obtaining a flat surface, sample grinding and finally, polishing.

### 2.2. Optic and Electronic Microscopy and Chemical Composition Evaluation

*Batches A* and *B* were subjected to optic and electronic microscopy to inspect the interface bone–adhesive–restoration material. Optical microscopy was performed using the Olympus SZX7. Higher magnifications of the interfaces were achieved by scanning electron microscopy (SEM), using an ESEM Philips XL 30 TMP. The chemical composition of the samples and the presence of the etching acid at the interface were evidenced by energy dispersive spectroscopy (EDAX).

### 2.3. FEA Simulation of Fracture Behavior

The structure, composition and macroscopic properties of adhesives change continuously from the cohesive to the adhesion zone, with the only region where the adhesive respects its intrinsic properties being the cohesive zone. As such, the local properties of the resulting composite will be different and directly dependent on the distance from the substrate to the adhesive. One of the frequent causes of interface separation is the diffusion of some of the adhesive components into the pores of the substrate, thus changing the local composition of the adhesive and therefore its properties.

To perform the fracture mechanics simulations, FEA of the assembly was conducted beforehand. A reconstructed molar was used as basic geometry, and using geometric operations, the adhesive layer and the restoration volume were designed, followed by component assembly. The elastic properties of the materials assigned to the individual components were extracted from the literature, namely the properties of dentin and enamel [29,30,31], the properties of adhesives [8,9,32,33] and the properties of the restorative material [13,34], and they are presented in Table 1.

The simulation was carried out in Ansys 2019. The discretization was conducted using tetrahedral elements for all components and constant sizing of 1 mm. The loading and fixing conditions simulated three physiological situations: pure vertical loading on the central pit surface (*Y*-axis); mesial 45° loading on the XoY plane, having the point of application in the central pit; and pure shear loading in the distal–mesial direction (*X*-axis). The fixed surface was as the bottom end of the model (Figure 3). The contact type between the components was bonding without permitting movements between interfaces.

The loading value was 150 N in one component for loading types A and C and two components of 106 N each (one vertical and one horizontal) for the type B loading. The point of application of the forces was the central groove surface in the case of the assembly model and the right–top surface of the simplified interface model. The values used represented average mastication forces [35,36,37] and were applied subsequently on the assembly model and on the simplified interface model.

The second numerical simulation was conducted using the Fracture-PreMeshed Crack module of Ansys 2019 to obtain the crack propagation and fracture behaviors. The loading directions and values were used to match the tooth simulation and were applied at the adhesion interface [38,39]. This method of subsequent modeling involves considering the initial complete model, solving it, extracting the area of interest and then carrying out the fracture mechanics simulation strictly on the isolated area, using a simplified model (Figure 4). The model contains the substrate (dentin), the adhesive and between them, crack initiation geometry. The length of the crack front was considered the thickness of the model, 1 mm, for all simulations, and the crack tip was at the center of the crack length. The mesh of the pre-existing crack was refined in the vicinity of the crack tip. The displacements used for crack propagation were in directions A, B and C and were used one at a time.

## 3. Results

### 3.1. Optic and Electronic Interface Evaluation and Chemical Composition

The images obtained by optical microscopy for the two analyzed groups are presented in the following paragraphs. First, the adhesion of All Bond 3, Bisco (ER 3-step) and Gradia Direct posterior to enamel and dentin is presented in Figure 5 and Figure 6.

Very good adhesion was observed between the adhesive system and the restoration material. The enamel prisms had slightly demineralized ends. The adhesive layer (Figure 5c-II) had a uniform thickness, and toward the junction with the enamel (Figure 5c-III), slight interpenetration was notable. At the adhesive–material interface (Figure 5c-II) of obturation, the connection was visible through the particles, which appeared modified in the first 20 μm of the thickness of the filling material.

The adhesive layer (Figure 6a,b-II) adhered very well to the restorative material and apparently to the dentin (Figure 6c-III). The adhesion was firm, and on the adhesive–dentin interface, uniform delimitation could be observed, marking the demineralization of the dentin following the adhesive in three steps.

The adhesion of Clearfil Universal Bond Quick, Kuraray Noritake Dental Inc. (1-step) and Gradia Direct posterior to enamel and dentin is presented in Figure 7 and Figure 8.

The adhesion achieved by the universal adhesive between the adhesive system and the restoration material was similar to the three-step adhesive. The enamel prisms no longer had demineralized ends, and no local demineralization could be observed. Although the adhesive layer had a uniform thickness, it failed to interpenetrate the dental tissue (Figure 7a,b-II).

In Figure 8, it can be seen that the uniform adhesive layer bound to the dentin through a much weaker hybridization due to reduced demineralization compared to the three-step adhesive. Fine fracture lines can be noted appearing in the dentin, probably from greater contraction of the adhesive in one step as opposed to three steps.

The investigation of adhesion to enamel and dentin was also evidenced by SEM microscopy at 100 and 500 X magnifications. The adhesion of All Bond 3, Bisco (ER 3-step) and Gradia Direct posterior, GC to enamel and dentin can be observed in Figure 9 and Figure 10. Here, the adhesive layer had a uniform thickness, filling the borders of the preparation with irregularities. The homogeneity of the adhesive and the interpenetration could be observed both at the adhesive–filling material junction and at the adhesive–enamel junction. Demineralization of the enamel as a separate step led to the increase in the contact surface of the dental tissue and the fusion of the adhesive with the enamel. Areas of hybridized complex can be observed at the adhesive–enamel interface (Figure 9a,b).

At the dentin level, the ends of the canaliculi were infiltrated with bonding from the adhesive three steps, and the adhesive layer had a uniform thickness and homogeneous structure without air gaps (Figure 10a,b).

The adhesion of Clearfil Universal Bond Quick, Kuraray Noritake Dental Inc. (SE 1-step) and Gradia Direct posterior to enamel and dentin is illustrated in Figure 11 and Figure 12 at magnifications of 100 and 500X, respectively.

At the enamel interface, there were microcracks that weakened the adhesion over time and represent weak points from which the fillings could detach, and infiltrations could occur. They appeared in all the tests performed with the Clearfill adhesive due to the higher shrinkage coefficient of the adhesive in one step. The adhesive layer was thinner at the enamel level and thicker at the dentin level, and the entire adhesive–dental tissue junction appeared as a crack, without fusing in any way with the enamel or dentin in any of the images obtained, indicating weaker adhesion (Figure 11a,b).

The adhesive was well delimited between the dentin and the obturation material and did not interfere with the dental tissue (Figure 12a,b). The adhesive in one step filled the imperfections obtained as a result of the cavity preparation but did not make a connection with the surface of the cavity. The picture indicates that this case is not as homogeneous as that with the three-step adhesive, leading us to the idea of weaker adhesion. The thickness of the adhesive layer was less than the case of universal adhesive and was non-uniform.

The composition evaluation (EDAX microanalysis) of the interface is presented in Figure 13 for batches A and B, respectively. In both interfaces, the presence of phosphorus can be identified, with higher peak in the case of batch A due to acid etching with phosphoric acid at a concentration of 37%. The demineralization step led to high spike of silicon ions and relatively reduced calcium ions. Batch B shows an even more reduced spike of calcium and silicon ions. The small quantities of calcium ions can decrease the strength of the superficial layer of dentin and implicitly create a potential weak point in the interface. Phosphorus occurs in smaller quantities, missing the acid demineralization step, but a small amount of fluorine, released from sodium fluoride contained in the universal adhesive, can be identified.

### 3.2. FEA Simulation of the Reconstructed Tooth

The simulation of the reconstructed molar model was carried out in a structural static module based on the geometric model and boundary conditions described in Section 2. The shear stress values were obtained for the entire assembly and in accordance with the orthogonal directions of the reference system. Figure 14 shows a cross-section of the model with three adhesion details, corresponding to the loading directions A, B and C, respectively. Each detail includes the tooth (left band), the adhesion layer (center) and the restoration material (right). Regardless of the loading direction, higher stresses can be observed (light blue) in the adhesive layer, confirming that the fracture propagation simulation should be addressed to this thin layer.

The full spectrum of in-plane shear stresses can be seen in Figure 15. Here, stresses that act only on the adhesive layer are presented as color images. It can be identified that areas of significant dimensional variation are the most likely to produce greater amounts of shear stress (warm colors).

### 3.3. FEA Simulation of Crack Propagation

Two sequences (initiation and final propagation) of crack growth are presented in the Figure 16 for loading scenarios A, B and C. The maximum shear stresses that arise in the vicinity of the crack tip can be observed. The progressive displacement of the fracture front is observed through the propagation of the refined mesh that follows the crack path. A very important aspect of the crack is the propagation path through the model. In the case of loading direction A, the failure followed the mode II fracture, proving the in-plane shear type of loading. In the real situation of this loading case, the dental adhesive would fracture oblique to the initial crack direction.

In the second fracture simulation (type B loading), the sequences showed a different direction of the crack propagation line (Figure 16B). It came closer to mode I due to the change in loading direction, and the most stressed areas were the crack tip and the insertion point of the load.

When horizontal loading was applied (type C loading), the stress could act on the longitudinal plane of the molar. By its nature, this load directly exposed the crack to opening and materialized a push stress on the restorative structure. Propagation occurs symmetrically, with a perfectly vertical crack collinear with the initial one (Figure 16, loading C). This propagation is pure mode I, caused by the opening of the flanks. The shear stresses are also uniformly distributed in the vicinity of the crack tip area, with values specific to the singularity phenomenon at the tip.

The stress intensity factor (SIF, *K*) is a parameter that determines the fracture toughness property of the material and is used to predict the stress state near the crack tip, when a remote load is applied to the structure. The magnitude of *K* depends on the specimen geometry, size and crack location, as well as by the loading value. The computation of *K* is obtained using Equation (1), where σ is the applied stress, a is the crack length, *W* is the specimen width, and *f*(*a*/*W*) is a geometry-dependent function [40,41]:(1)K=σπ·a·faW

The values of stress intensity factors (SIFs) were determined for all loading directions and for the three possible failure modes (I, II and III). The crack propagation front for the three modes is shown in Figure 17, where blue represents low values of *K*, while red represents high values of *K*. The stress intensity factor has the highest value in the middle region of the crack front for mode I failure and the lowest value for mode III, when the vertical loading direction (A) is applied.

By analyzing the SIFs values extracted from the crack fonts of all simulation scenarios and comparing them with the fracture toughness of the adhesives, the probability of interface failure for a certain combination can be determined in the material–geometry–loading direction. The stress intensity factors are depicted in Table 2 for both types of adhesives and for all loading directions. The *K* values denoted as high correspond to the fourth-generation adhesive Allbond3, which has a considered mode I fracture toughness of 1.52 MPa·m^1/2^, while low denotes the SIFs of the universal adhesive Clearfil, with a considered mode I fracture toughness of 0.62 MPa·m^1/2^ [22,23,24,25,28].

The negative values of SIFs in the Table 2 are determined for compressive stress fields acting on the crack front. All in all, the values without signs were compared to the fracture toughness of each adhesive to determine the probability of the interface to resist fracture propagation. As can be observed, in the majority of cases, the SIFs highly exceed the fracture toughness values, so the susceptibility of failure by crack propagation is very high.

## 4. Discussions

In the present study, the stresses acting at the dentin–adhesive–restoration material interface of an onlay restored molar were determined by numerical simulation. The stress values obtained in this simulation correspond to elastic displacement of the interface. The results of this first simulation were used to establish the loading conditions of the simplified crack propagation model. Independent of loading direction, higher stresses were determined in the adhesive layer, confirming that the fracture propagation simulation should be addressed to this thin layer.

The fracture mechanics simulation of an adhesive provides new information about the behavior of interfaces when cracks occur in the superficial, marginal layer. The results obtained were then compared with the fracture toughness of the materials to determine fracture susceptibility.

The SIFs at the crack front had the highest values for mixed mode loading (B), when the load was placed 45° obliquely to the initial crack. In pure mode I loading (A), the SIF values were also very high for both adhesives (KIhigh and KIlow). The overall observation is that crack opening is more likely to produce fracture propagation in dental adhesives than in-plane shear, which can be applied from a vertical load.

Comparing the two adhesives, the fourth-generation adhesive had a considerable higher mode I fracture toughness than the universal adhesive (about 40% higher) and only 30% higher stress intensity factors recorded for the considered conditions. Therefore, in the given conditions, the three-step adhesive will better perform.

The SEM investigation confirmed firm adhesion of dental adhesives to the dental tissue and the presence of a well-represented hybridized layer. The interpenetration of the adhesive layer with the dental one can be observed in Figure 9b.

Adhesion is firm to dentin and enamel with the All bond 3 adhesive in the presence of acid gavage in a separate step. Micro-retentiveness can be observed, as well as a slightly crenellated surface area after the acid etching, so the mechanical adhesion is increased compared to the case in which there is no acid etching (Figure 9 and Figure 10).

When using the universal adhesive, we observed slightly poor adhesion of the adhesive, with well-defined interface areas and without fusing (Figure 11 and Figure 12). The higher shrinkage of the universal adhesive due to the presence of the HEMA component may led to crack formation, and since there is no possibility of observing the issue during clinical treatment, the lather obturation can fail (Figure 12b). The dental tissue at the adhesion interface is smooth and regular without providing mechanical retentiveness.

The thickness of the universal adhesive layer is relatively smaller and shows reduced hybridization, while the layer thickness of the fourth-generation adhesive is greater and can repair some imperfections of the cavity preparation stage. It also stands out with greater hybridization. The thickness of the layer is approximately constant for the fourth-generation adhesive, while important thickness variations can be observed for the universal adhesive.

The study conducted by Jang JH et al. comparing the Clearfil and Allbond adhesives revealed that the universal adhesive, despite the simplified use technique, presented good adhesion to the dental hard tissues, with slight deficiencies in the case of atypical cavities [42]. Chowdhury AFMA et al., following an SEM comparison of self-etch and multi-etch adhesives, obtained similar results to ours. The fewer steps that the adhesives protocol had, the more ineffective they became [43]. Additionally, the study by Kasahara Y et al. concluded that three-step adhesives are superior to universal and two-step adhesives [44]. Using SEM analysis, Kanniappan G et al., in their comparative study of adhesive interface–dental structures made with universal adhesives and total-etch systems using the CLSM technique, demonstrated the superiority of adhesives that use the total-etch work protocol [45]. In an enamel bonding strategies study, the authors concluded after tests performed on fillings with universal adhesives and three-time adhesives that the universal adhesive reduced the working time, but the long-term enamel adhesion performance was inferior to the adhesives in two or three steps, and the appearance of infiltrations and later of secondary carious lesions was favored [46].

For the three-step adhesive, the EDAX analysis showed an appropriate balance between the calcium ions remaining in the structure, which provide strength to the dental tissue, and the free ones. It is also possible to see the inversion spike between Ca and Si in the demineralized superficial layer, which plays an important role in reducing post-obturation sensitivity. Similar changes in acid demineralization were also found by other authors who considered changing the Ca/Si ratio to be beneficial because the new bonds formed after the dissolution of the superficial layer of dentin can block the dentinal canaliculi and decrease any post-obturation sensitivity [47]. Another study revealed that, following the demineralization step, the released calcium ions can bind to 10-MDP monomers from the adhesive composition, forming stable 10-MDP-Ca bonds that increase adhesion and reduce the risk of occurrence of marginal infiltrations [48].

## 5. Conclusions

This paper presented a comparative study of the interface fracture behavior between fourth-generation three-step adhesive and a universal adhesive. The study relied on optic and SEM investigation of the dentin–adhesive–restoration material interface and also on FEA simulation of fractures. The microscopic qualitative analysis revealed inferior adhesion of the universal adhesive to the surrounding substrates. The layer thickness of the universal adhesive is small and variable, while the layer thickness of the fourth-generation adhesive is greater and approximately constant. Hybridization is clearly superior in the case of the fourth-generation adhesive, and the interface between the dental tissue and the adhesive is slightly crenulated, increasing the contact surface. The large contact surface and the lack of air contact gaps allowed the three-step adhesive to attain much better marginal adaptation compared to the universal adhesive. The simulations showed in comparison that the fourth-generation adhesive has a considerable higher mode I fracture toughness than universal adhesive (about 40% higher) and only 30% higher SIFs recorded. Therefore, in the given conditions, the fourth-generation adhesive will better perform in terms of preventing crack formation and propagation.

## Figures and Tables

**Figure 1 materials-16-05834-f001:**
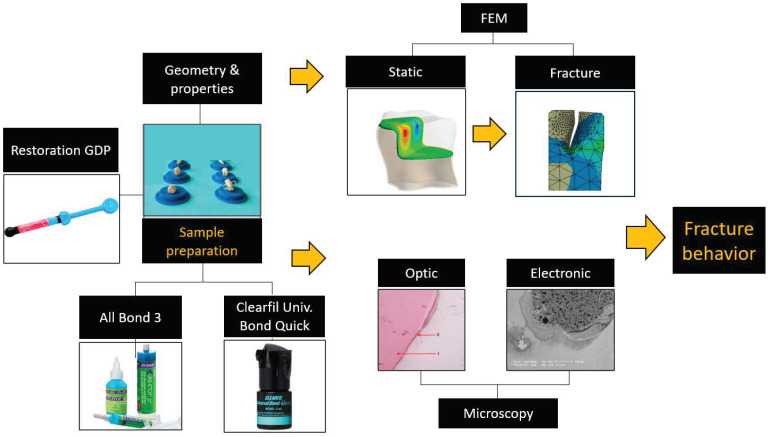
Illustration of the investigation protocol.

**Figure 2 materials-16-05834-f002:**
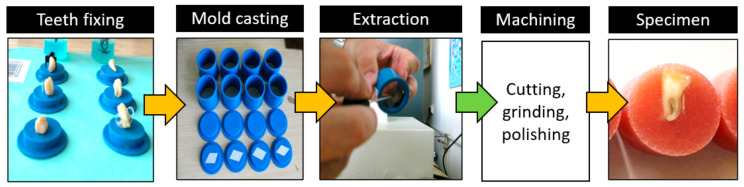
Steps in sample preparation.

**Figure 3 materials-16-05834-f003:**
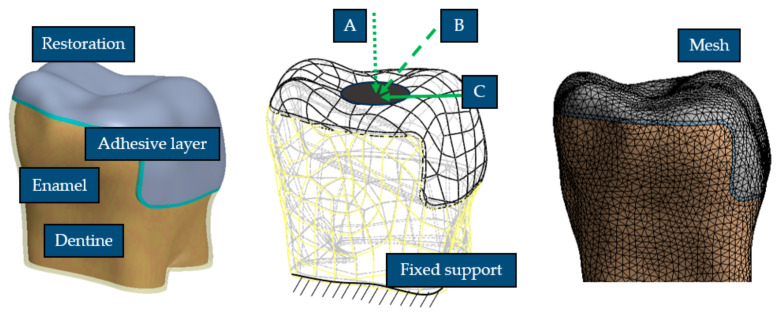
Geometry, boundary conditions and mesh.

**Figure 4 materials-16-05834-f004:**
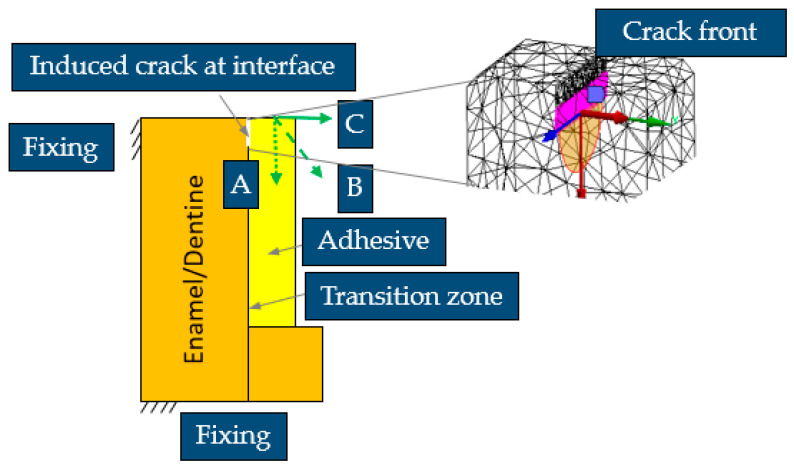
Simplified interface model and refined mesh at crack tip.

**Figure 5 materials-16-05834-f005:**
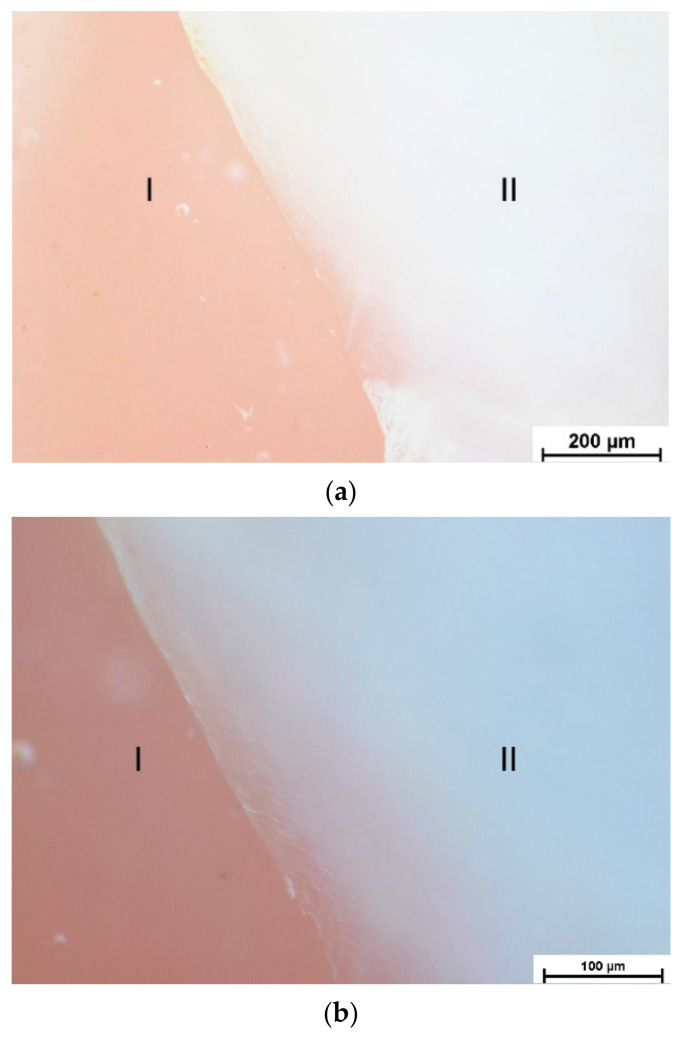
Restoration–filling material–adhesive–enamel interface, (**a**) 200 µm scale; (**b**) 100 µm scale; (**c**) 20 µm scale.

**Figure 6 materials-16-05834-f006:**
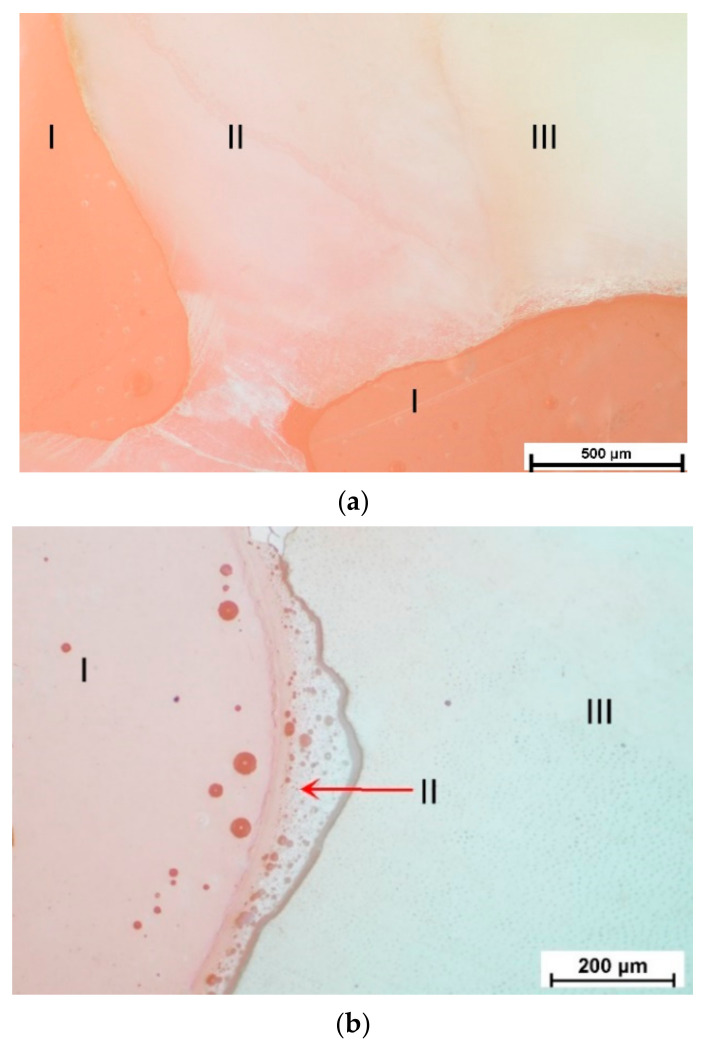
Restoration–obturation material–adhesive–dentin interface, (**a**) 500 µm scale; (**b**) 200 µm scale; (**c**) 100 µm scale.

**Figure 7 materials-16-05834-f007:**
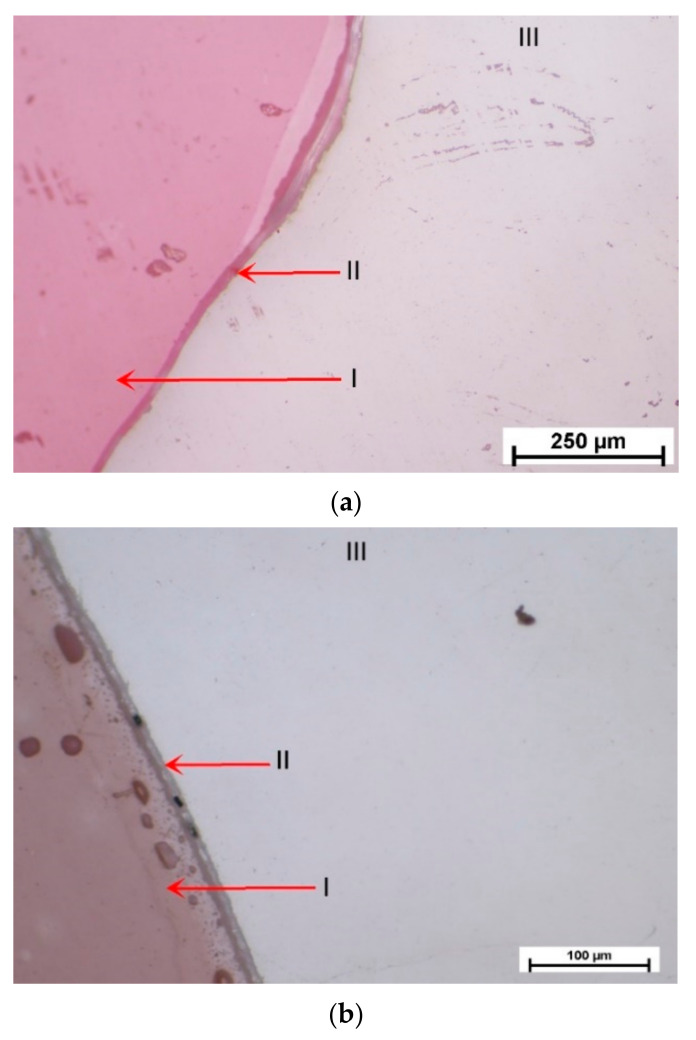
Interface restoration–adhesive–enamel, (**a**) 250 µm scale; (**b**) 100 µm scale.

**Figure 8 materials-16-05834-f008:**
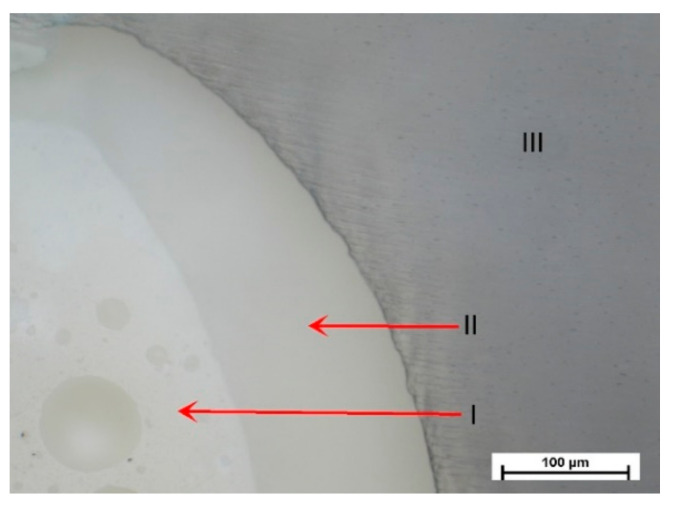
Interface restoration–adhesive–dentine, 100 µm scale.

**Figure 9 materials-16-05834-f009:**
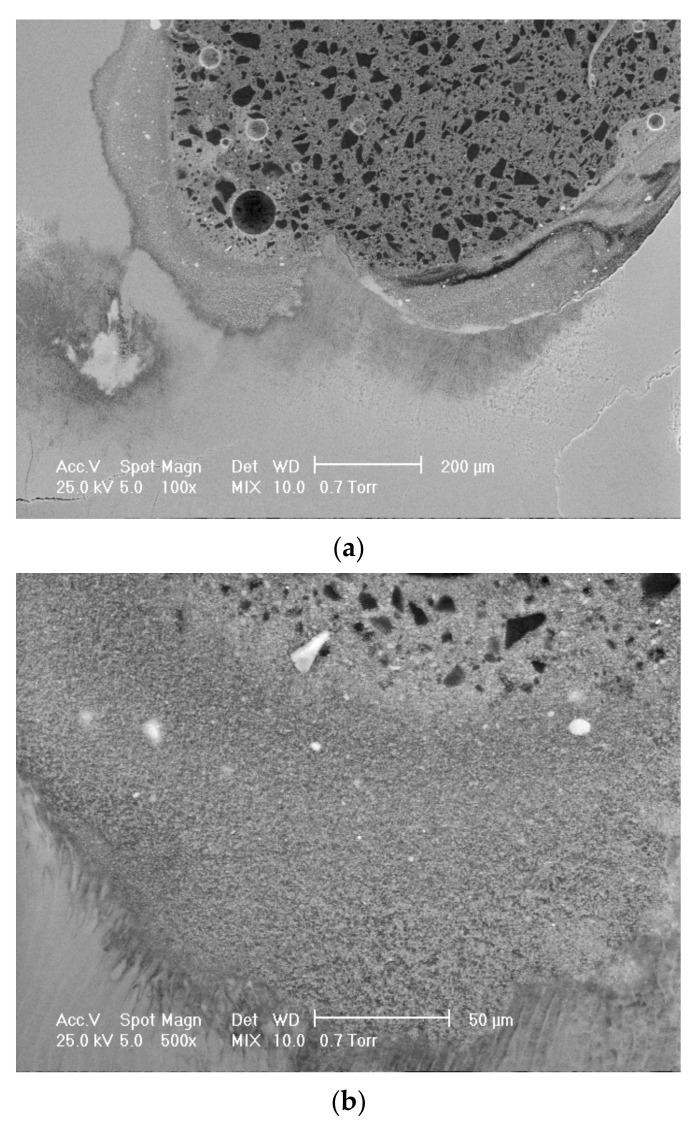
Interface restoration–adhesive–enamel, SEM. (**a**) 100X magnification; (**b**) 500X magnification.

**Figure 10 materials-16-05834-f010:**
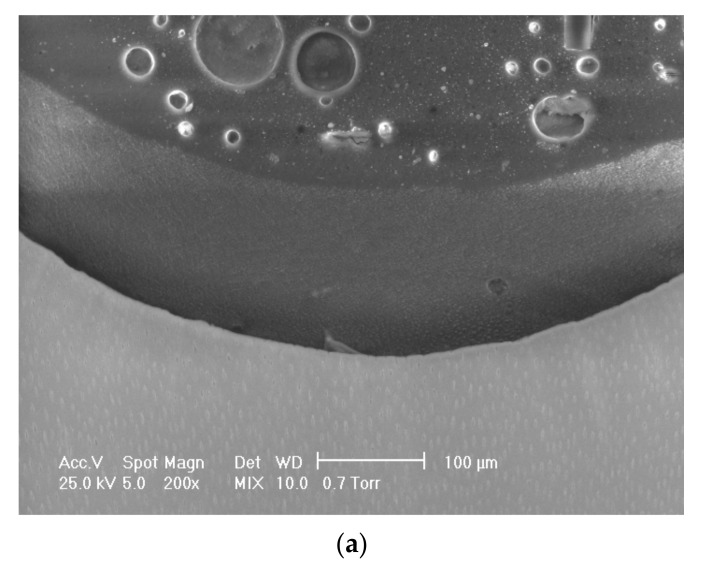
Interface restoration–adhesive–dentin, SEM images. (**a**) 200X magnification; (**b**) 500X magnification.

**Figure 11 materials-16-05834-f011:**
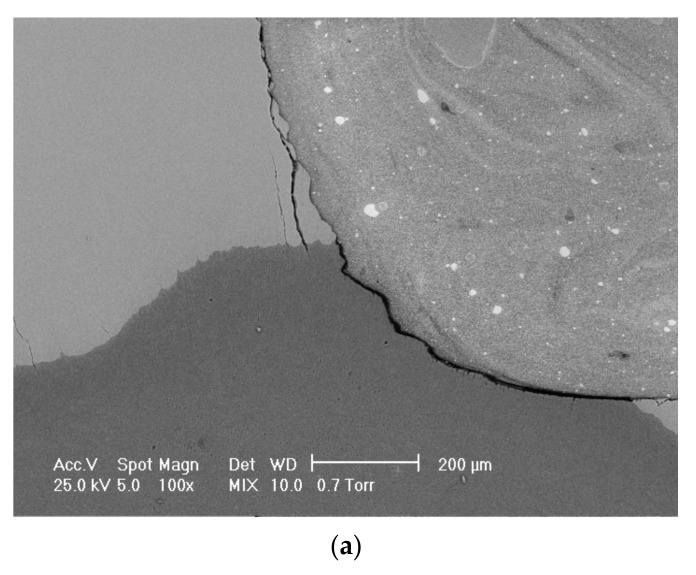
Restoration interface–adhesive–enamel, SEM images. (**a**) 100 magnification; (**b**) 200X magnification.

**Figure 12 materials-16-05834-f012:**
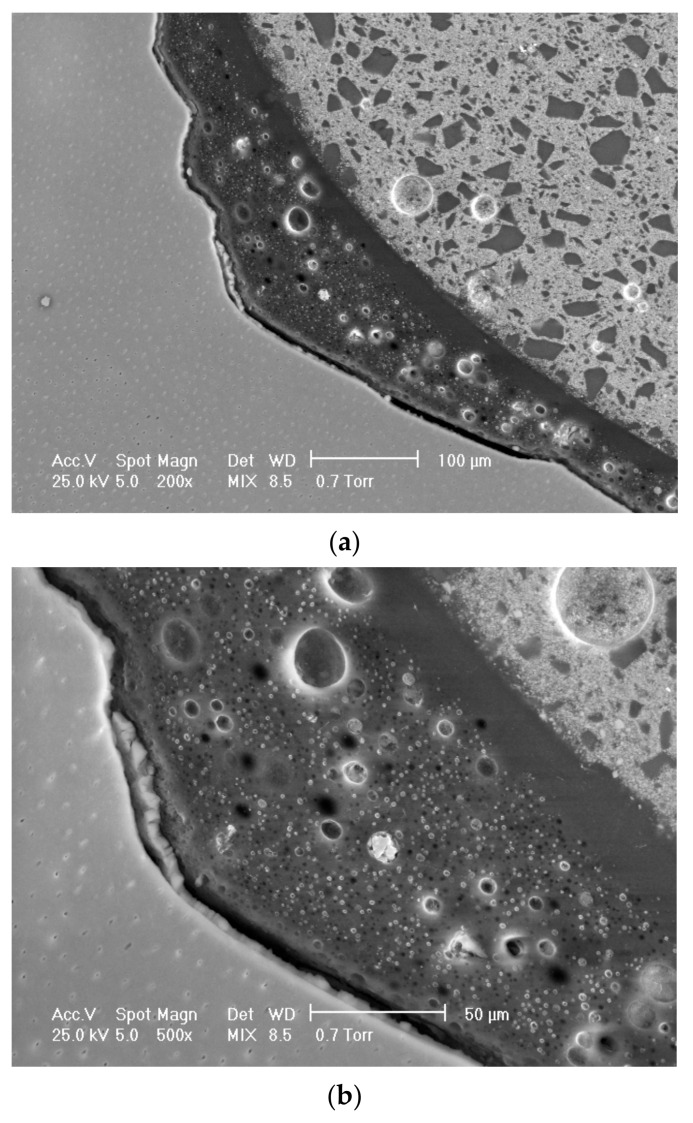
Restoration interface–adhesive–dentine, SEM images. (**a**) 200X magnification; (**b**) 500X magnification.

**Figure 13 materials-16-05834-f013:**
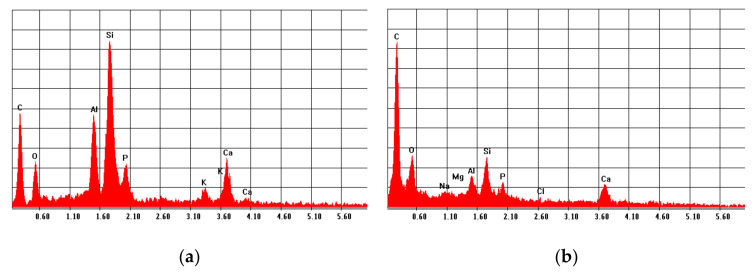
EDAX compositional results for adhesive interface of (**a**) batch A and (**b**) batch B.

**Figure 14 materials-16-05834-f014:**
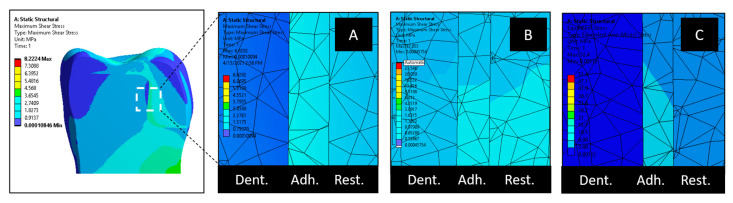
Shear stresses in transversal section of the reconstructed molar, according to loading directions: vertical (**A**), oblique (**B**) and horizontal (**C**).

**Figure 15 materials-16-05834-f015:**
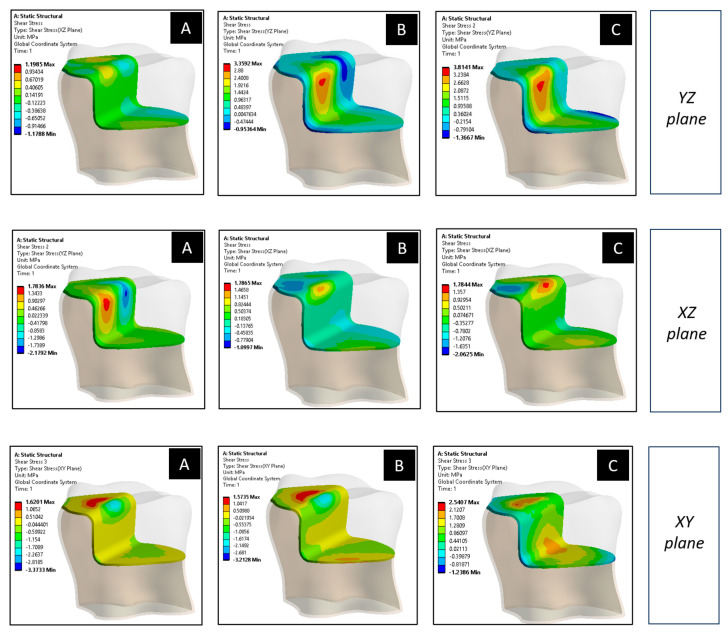
Shear stresses in adhesive layer according to loading directions: vertical (**A**), oblique (**B**) and horizontal (**C**).

**Figure 16 materials-16-05834-f016:**
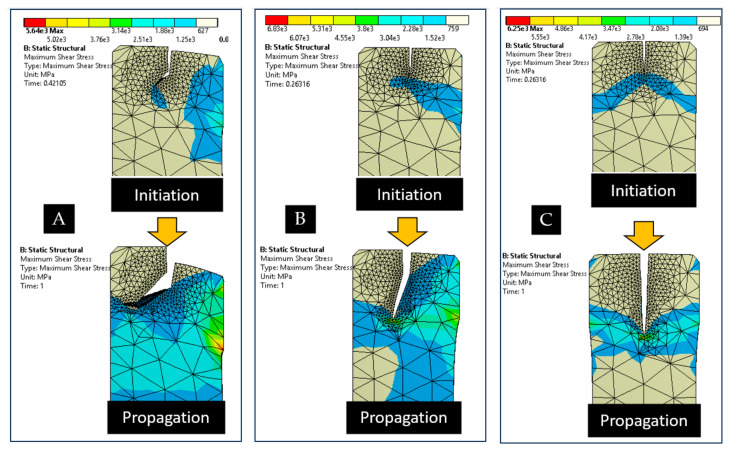
Crack initiation and propagation according to the loading direction: vertical (**A**), oblique (**B**) and horizontal (**C**).

**Figure 17 materials-16-05834-f017:**
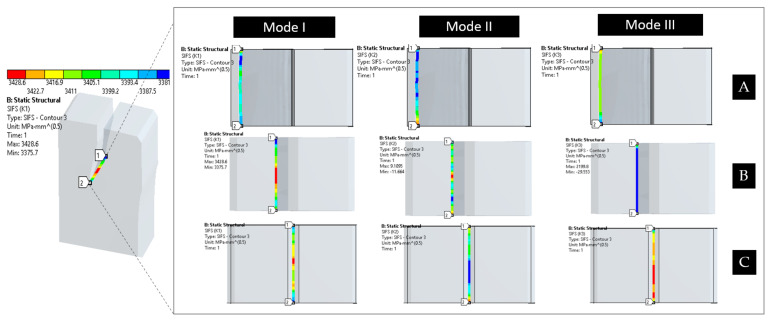
Stress intensity factors in fracture modes I, II and III, along the crack front line and according to the loadings (**A**–**C**).

**Table 1 materials-16-05834-t001:** Mechanical properties used in simulation.

Structure/Material	Density [kg/m^3^]	Young Modulus [GPa]	Poisson Ratio [-]	Fracture Toughness [MPa·m12 ]
Dentin	2000	17.0	0.30	-
Enamel	2750	74.0	0.23	-
Restoration material	2480	8.0	0.34	0.38
Universal adhesive	2400	7.7	0.24	0.62
Fourth-generation adhesive	2400	8.1	0.25	1.52

**Table 2 materials-16-05834-t002:** SIF values recorded for both adhesives in A, B and C loading directions.

K [MPa·m12 ]		$K_{I}^{high}$	$K_{II}^{high}$	$K_{III}^{high}$	$K_{I}^{low}$	$K_{II}^{low}$	$K_{III}^{low}$
A	max.	30.46	−0.69	11.68	8.80	−0.15	4.69
	min.	29.92	−1.02	−0.29	8.58	−0.29	−6.28
B	max.	103.20	0.24	36.19	30.56	0.12	9.28
	min.	102.59	−0.58	−0.90	29.67	−0.43	−9.89
C	max.	95.53	0.34	1.45	28.11	0.12	0.63
	min.	93.58	−1.31	−0.43	27.41	−0.08	−0.66

## Data Availability

The data presented in this study are available on request, from the corresponding author.

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
