# Peer review of "Comparative Study on Interface Fracture of 4th Generation 3-Steps Adhesive and 7th Generation Universal Adhesive"

_materials, 2023, doi:10.3390/ma16175834_

Round 1
Reviewer 1 Report
This manuscript titled Comparative Study of Interface Fracture Behavior between 4th Generation and Universal Dental Adhesives, presents two approaches to the problem, one from ex vivo specimens, and another from a modeling analysis.
This has some perspective problems, beginning with that, the 3-step generation adhesives are now not commonly used by clinicians, because of the development of new materials which reduce procedure times. Up to now the most effective adhesive system in terms of mechanical behavior at the interface of restoration is the 5th generation of adhesives, regardless of the well-reported advantages of new generations of adhesives (6th and 7th).
Another drawback of this research is that for the in vitro specimens, after the microscopy there is no quantitative analysis between the interface of the materials, in the text named “batches”. Also, there is no statistical analysis performed.
For the case of the simulation analysis, the main problem is that the restorations did not follow the same shape and morphology as the in vitro specimens. Making this comparison ineffective.
In the abstract's conclusions the sentence “Significant differences were noted between adhesives” has no supporting data in the manuscript.
There are some relevant facts in the Introduction that are not properly referenced.
In Table 1. The density units are incorrect because the 3 should be super indexed. And for the numbers of the modulus, the decimal separator should be a period “.” not a comma “,”.
In the results, line 290, refers to a Ca/Si ratio obtained by EDX, but there is no data about that ratio. Then, the authors compare EDX elemental composition of adhesion interfaces of batch 1 and batch 2 but only is shown the data of batch 1. Also, all numerical results (at.% or wt.%) from the EDX analysis of each reported element from different adhesion interfaces are not reported.
In Figure 5b the scale bar is missing.
A final recommendation is that all claims stated in the discussion and conclusions section must be clearly supported by the data presented in the results.
English must be revised, because of some issues detected.
Line 148- The word "Th".
Line 159- Revise the term "study plan".
Line 186- The word "Tha" is incorrect.
Author Response
Dear reviewer, please find the attached file containing the answers to your questions. Thank you

Reviewer 2 Report
1. Abstract should be in a single paragraph.
2. Lack of dental composite materials description in the introduction section. Authors to advice to add recommended articles into introduction section for describing the restorative materials
a. A comprehensive review: Physical, mechanical, and tribological characterization of dental resin composite materials
b. Investigation of physical, mechanical, thermal, and tribological characterization of tricalcium phosphate and zirconia particulate reinforced dental resin composite materials
3. Materials and methods section: Details of restorative composite materials is missing. Kindly add
4. Section 2.1. Specimen fabrication: "The cold incorporation of the samples was made in epoxy resin that........." Does epoxy resin significant for restorative materials. Epoxy resins have toxic nature.
5. Did authors performs experimental work on fracture toughness.
6. Load condition in FEM section is missing. Kindly add
7. Did authors finds the displacement in FEM.
8. Figure 16; why did crack propagation in A higher than others.
9. Conclusion should be in a single paragraph.
Author Response
Dear reviewer, please find the attached file were we addressed all your comments.
Thank you

Round 2
Reviewer 1 Report
The authors did great work fixing manuscript issues, had perform needed improvements, and provided reasonable responses and explanations to this reviewer's queries. The manuscript is now notoriously improved.
Reviewer 2 Report
Lack of novelty in this work. Many other factors should be considered for the analysis of stress and strain.